# Glutathione Non-Covalent Binding Sites on Hemoglobin and Major Glutathionylation Target betaCys93 Are Conservative among Both Hypoxia-Sensitive and Hypoxia-Tolerant Mammal Species

**DOI:** 10.3390/ijms25010053

**Published:** 2023-12-19

**Authors:** Anastasia A. Anashkina, Sergey Yu. Simonenko, Yuriy L. Orlov, Irina Yu. Petrushanko

**Affiliations:** 1Engelhardt Institute of Molecular Biology, Russian Academy of Sciences, 32 Vavilova Str., 119991 Moscow, Russia; irina-pva@mail.ru; 2Biotechnology Division, Research Center for Translational Medicine, Sirius University of Science and Technology, 1 Olympic Ave., 354340 Sirius, Russia; s.simonenko.bio@gmail.com; 3Digital Health Institute, I.M. Sechenov First Moscow State Medical University of the Ministry of Health of the Russian Federation (Sechenov University), 8-2 Trubetskaya Str., 119991 Moscow, Russia; y.orlov@sechenov.ru

**Keywords:** hypoxia, hemoglobin, glutathione, redox sensitivity, hypoxia-tolerant species, alpha subunit, beta subunit, cysteine residues

## Abstract

Intracellular tripeptide glutathione is an important agent of cell survival under hypoxia. Glutathione covalently binds to SH groups of hemoglobin cysteine residues, protecting them from irreversible oxidation, and changes its affinity to oxygen. Reduced glutathione (GSH) can also form a noncovalent complex with hemoglobin. Previously, we showed that hemoglobin tetramer has four noncovalent binding sites of glutathione GSH molecules inside, two of which are released during hemoglobin transition to deoxy form. In this study, we characterized the conserved cysteine residues and residues of noncovalent glutathione binding sites in the sequences of a number of hypoxia-tolerant and hypoxia-sensitive mammals. The solvent accessibility of all HbA and HbB residues in oxy and deoxy forms was analyzed. The alpha subunit of all species considered was shown to have no conserved cysteines, whereas the beta subunit contains Cys93 residue, which is conserved across species and whose glutathionylation changes the affinity of hemoglobin for oxygen 5–6-fold. It was found that the key residues of noncovalent glutathione binding sites in both alpha and beta subunits are absolutely conserved in all species considered, suggesting a common mechanism of hemoglobin redox regulation for both hypoxia-sensitive and hypoxia-tolerant mammals.

## 1. Introduction

Erythrocytes are hemoglobin-containing cells which can bind O_2_ with hemes—Fe^2+^—containing prosthetic groups in lungs and release it in peripheral tissues [1]. Hemoglobin is a well-studied protein having quaternary structure including two pairs of identical subunits.

Subunits of human hemoglobin (*Homo Sapience*^N^) (N is the notation for the normoxy species) are encoded by genes: HBA1 and HBA2 (α-subunit, P69905), HBB (β-subunit, P69905), HBG1 and HBG2 (γ-subunit), HBD (δ-subunit, P02042), HBE1 (ε-subunit, P02100), HBZ (ζ-subunit, P02008), and HBQ1 (θ-subunit, P09105) (2022). Expression of these genes is the most significant in the bone marrow, the spleen, placenta and other female reproductive tissues (HBD, HBG1, and HBG2 only) or male reproductive tissues (HBZ only). Lower expression levels of these genes can be detected in liver, lungs, cerebellum and cerebral cortex tissues [2]. Expression and biosynthesis levels of different hemoglobin subunits depends on age and are probably regulated epigenetically [3].

Hemoglobin is very densely packed in erythrocytes and is surrounded by a thin layer of bound water [4]. A high concentration of hemoglobin stabilizes its tetrameric quaternary structure [5]. The binding and releasing of oxygen are cooperative due to the Bohr effect. In different stages of development and in different species, hemoglobin binding affinity for O_2_ and oxygen release in tissues may differ. Most animal species live and reproduce themselves at a normal level of atmospheric oxygen, about 20%. When it decreases, oxidative stress occurs in tissues and it leads to oxidative modifications of proteins and free-radical damage of cell compartments. However, some animal species are resistant to oxidative stress caused by lack of oxygen in their natural environment. One of the widely known ones is the naked mole rat *Heterocephalus glaber*^HR^ (HR is the notation for the hypoxia-resistant species), which demonstrates exceptional resistance to hypoxemia (~5% O_2_) [6,7]. These animals can effectively reproduce themselves even at low oxygen concentrations.

There are several identified mechanisms of hypoxia resistance in mammals [6,8]. For example, increased blood vessel density allows Spalax to tolerate underground hypoxia [9]. Both humans [10] and animals [11,12] living in hypoxic conditions may have elevated hematocrit, hypoxia-inducible factor-1alpha (HIF-1α), erythropoietin and hemoglobin levels. In humans living on the Tibetan plateau, hemoglobin concentration is independent of altitude but it rises with altitude in organisms living on the Andean highland [10]. In the Ethiopian highlands, both hemoglobin concentration and arterial oxygen saturation are independent of altitude [13,14]. This means that hemoglobin concentration and arterial oxygen saturation may be an inherited genetic adaptation. Oxygen affinity and saturation are closely related to the properties of hemoglobin molecules in animal species. It is differences in the sequence and, consequently, the structure of hemoglobin that are responsible for differences in the affinity of oxygen binding in the lungs and the efficiency of its release in body tissues.

In recent studies [15,16], we have shown that hemoglobin tetramers in the oxy form are depots of glutathione, which is partially released during the transition to the deoxy form and protects tissues under hypoxia. The formation of a noncovalent complex of hemoglobin with glutathione changes the properties of hemoglobin and leads to a slight increase in affinity to oxygen [15,16]. Based on our experimental data and molecular docking data, we hypothesized that there are two pairs of glutathione binding sites with different affinities in the central cavity of the oxyhemoglobin tetramer. The high affinity pair of binding sites is located at the interface between α- and β-hemoglobin chains, and the second pair of sites with slightly lower affinity is located between the two β-chains [16]. Under deoxygenation, two GSH localize in the “pocket” between two beta-globin chains release [16]. In this study, we characterized the conservation of amino acid residues forming all four GSH binding sites in some hypoxia-tolerant and hypoxia-sensitive species.

There is evidence that glutathionylation of the Cys93 of HbB leads to a 5–6-fold increase in oxygen affinity [17]. We have shown that Cys93 of HbB in adult hemoglobin is solvent-accessible only in deoxy form, residue Cys112 of HbB is accessible in both oxy and deoxy forms with small ASA area, and Cys104 of HbA is unexcessively inaccessible and its glutathionylation has not been confirmed [15,16]. Oxidative stress can lead to the oxidation of GSH to GSSG and SH groups of proteins to SOH and further to SO_2_ and SO_3_ (irreversible oxidation). The reduced form of glutathione interacts with SOH group of protein to form a disulfide bond (proteinS-SG) and a water molecule or oxidized glutathione interacts with the SH group of protein, resulting in protein glutathionylation [18]. Such modifications modulate the function of the protein, affecting its activity. Under normal conditions, glutaredoxin removes glutathionylation from cysteine residues, restoring the original function of the protein. Glutathionylated hemoglobin is the marker of oxidative stress of red blood cells [19]. Here, we tested the hypothesis that the number of hemoglobin cysteines available to the solvent correlate with the severity of oxidative stress conditions to which this species is adapted.

Hypoxia-tolerant species have a higher affinity of hemoglobin for oxygen and a higher nitrite reductase activity than sensitive species [20,21]. The neotenic theory of hypoxia tolerance suggests that some hypoxia-tolerant animals use isoforms of hemoglobin that are similar to that of neonatal mammals [7]. This assumption looks logical because fetal and embryonic hemoglobins have greater binding affinity. We decided to see what place in the hemoglobin sequence hierarchy the hemoglobin sequences of hypoxemia-resistant mammals occupy and to test this hypothesis.

## 2. Results

### 2.1. Hemoglobin Hierarhy

We found references to hypoxia-resistant mammals in the literature and compiled a list of hemoglobins of these species (Appendix A).

Looking at the hierarchical tree of hemoglobin sequences of the selected mammalian species (Appendix A), it can be seen that the alpha-like sequences, with two exceptions, are well classified. Only the Hemoglobin X sequence A7M7S6 of *Mus musculus*^N^ (Mouse), which fell into the zeta hemoglobin cluster, and the alpha-chain sequence A0A8I6AI13 of *Rattus norvegicus*^N^ (Rat), which fell into the theta chain cluster, are out of the overall harmonious picture.

At first glance, beta-like sequences do not have such a clear separation of chain types, but this is an artefact of the clustering and tree drawing algorithm (Appendix A). Two clusters of beta-chains could well be placed next to each other, and then two clusters of epsilon chains would be next to each other.

A number of conclusions can be drawn from this figure. For example, the gamma chain of hemoglobin J7LGZ1 of *Pantholops hodgsonii*^HR^ is closest to fetal hemoglobin beta P02082 of *Capra hircus*^HR^ and fetal hemoglobin beta P02081 of *Bos taurus*^HR^. The delta chains of *Homo sapiens*^N^ (P02042), *Rhinopithecus roxellana*^HR^ (A0A2K6P3G0) and *Theropithecus gelada*^HR^ (A0A8D2K6V2) were found to be more similar to the beta-hemoglobins of different species.

Considering the hypothesis of neonatal origin of hemoglobins of hypoxia-resistant species, we can conclude that the sequences of both beta-chains of aquatic mammals *Hydrodamalis gigas*^HR^ (A0A481WPM8, A0A481WR13) and the single HbB isoform of *Trichechus manatus latirostris*^HR^ (A0A2Y9ECT2) are indeed closer to delta chains of other species. Also, one of the four HbB isoforms A0A8C6RUI9 of *Nannospalax galili*^HR^ is more similar to epsilon hemoglobins. Two of the six HbB isoforms of *Mus musculus*^N^ (P04443, P04444) are much more similar to epsilon hemoglobins, e.g., epsilon hemoglobins A0A452F723 of *Capra hircus*^N^ and A0A6I9I4T5 of *Vicugna pacos*^HR^, than to beta-hemoglobins.

Some Uniprot classifications of sequences look wrong. For example, theta hemoglobin P04246 of *Sus scrofa*^N^ looks like epsilon hemoglobins.

The most dissimilar sequences are the hemoglobin sequences zeta (A0A8C2R2Z3) of *Capra hircus*^N^ and alpha (G5BXY4) of *Heterocephalus glaber*^HR^ (two the lowest branches of the phylogenetic tree, Appendix A).

### 2.2. Cysteine Residues in HbA and HbB Sequences

#### 2.2.1. HbA

Interestingly, all alpha hemoglobin isoforms of *Bos taurus*^N^ (P01966) and *Bos mutus grunniens*^HR^ (P01968, P01967) contain no cysteine at all (Appendix A).

Most HbA sequences have only one cysteine residue at position 104 like in humans. But all HbA isoforms of *Pantholops hodgsonii*^HR^ (F6MFG2 and Q0ZA50), *Capra caucasica caucasica*^HR^ (A0A7D5Y015), *Capra hircus cretica*^HR^ (A0A7D5Y174), *Capra nubiana*^HR^ (A0A7D5Y1A3) and *Capra hircus*^N^ (P0CH25, P0CH26) have only one cysteine residue at position 111 not 104 (Appendix A).

Alpha hemoglobin sequences of *Ailurus fulgens*^HR^ (P18969) and *Talpa europaea*^N^ (P01951) contain two cysteines at both positions 104 and 111. *Rattus norvegicus*^N^ alpha hemoglobin isoform A0A8I6GMB4 contains two cysteines at positions 13 and 104. At very different positions, 9 and 25, *Heterocephalus glaber*^HR^ G5BXY2 sequence contains two cysteines.

Two of four probably functional isoforms of *Rattus norvegicus*^N^ (A0A1K0FUB4, P01946) contain three cysteines at positions 13, 104 and 111. Both alpha hemoglobin isoforms of *Tamias umbrinus*^HR^ (T1YT26 and T1YVM3) contain three cysteines at 13, 104 and 111 positions.

So, positions 13, 104 and 111 are common for the most cysteine residues of HbA, and *Heterocephalus glaber*^HR^ G5BXY2 sequence contains only two cysteines at positions 9 and 25. Of these residues, Cys104 and Cys25 are not solvent-accessible (Appendix A) both in oxy and deoxy forms; the remaining residues are solvent-accessible and can theoretically undergo oxidation and glutathionylation. Cysteine 104 of the alpha subunit, in both oxy and deoxy forms, contacts with Gln127 of the beta subunit, so we believe it is important for maintaining the stability of the alpha/beta subunit interface. β Cys13 in mouse HbbD beta globins was more susceptible to disulfide exchange with GSSG than βCys93 [22] by *in vitro* studies. Also, it is known that βCys13 is able to bind dimethylarsinous acid in rat blood [23]. The function and role of *Heterocephalus glaber*^HR^ G5BXY2 hemoglobin cysteine residues have not been studied to date.

#### 2.2.2. HbB

All organisms (hypoxia sensitive and hypoxia resistant) have at least one cysteine at position 93 in hemoglobin beta-chains (Appendix A).

At positions 93 and 112, there are cysteines in the sequences of three beta-hemoglobin groups: (1) hominidae and bears *Rhinopithecus roxellana*^HR^ (A0A2K6Q8D1), *Homo sapiens*^N^ (P68871), *Theropithecus gelada*^HR^ (A0A8D2FZZ9 and P02029), *Ailurus fulgens*^HR^ (P18982) and *Ursus thibetanus*^HR^ (P68012); (2) European hedgehog and mole *Erinaceus europaeus*^N^ (A0A1S3AE42, A0A1S3WPY1 and P02059) and *Talpa europaea*^N^ (P02061); (3) sealiforms *Hydrodamalis gigas*^HR^ (A0A481WR13 and A0A481WPM8), *Dugong dugon*^HR^ (A0A481WQB0) and *Trichechus manatus latirostris*^HR^ (A0A2Y9ECT2).

Two cysteines at positions 93 and 125 appeared in two groups of beta-hemoglobin sequences: five of the eight potential isoforms of *Rattus norvegicus*^N^ (A0A0G2JTW9, P02091, A0A8I5ZXF1, A0A8L2R1Q6 and A0A8L2R6L7) and both isoforms of *Cavia porcellus*^HR^ (P02095 and A0A7T7JJ63).

Positions 13 and 93 locate cysteines in two of the five potentially functional sequences of *Mus musculus*^N^ (P02088 and P02089) beta-hemoglobins.

One of the four sequences of *Nannospalax galili*^HR^ HbB (A0A8C6RII4) is the most highly divergent. It contains cysteines at positions 13, 32, 50 and 93.

In the beta-chain, Cys93 and Cys32 residues are solvent-accessible only in deoxy form, and Cys112 is solvent-accessible in either oxy or deoxy forms (Appendix A).

The spatial arrangement of all cysteine residues mentioned in Section 2.2.1 and Section 2.2.2 is shown in Figure 1.

### 2.3. GSH Binding Pockets Conservativity

In our recent work, we found that human oxyhemoglobin can bind four glutathiones and identified four GSH binding sites inside the adult oxyhemoglobin tetramer of *Homo sapience*^N^ using molecular modeling [16]. Two symmetric sites for oxy- and deoxyhemogemobin include hemoglobin alpha-chain residues Val1, Leu2, Pro95, Val96, Phe98, Lys99, Ser102, His103, Leu106, Asp126, Lys127, Leu129, Ala130, Ser131, Ser133, Thr134, Thr137, Ser138 and Tyr140, among which residues Leu2, Pro95, His103, Leu106, Asp126 and Lys127 are absolutely conserved in all organisms, both normoxia and hypoxia-resistant organisms, and residues of beta-chains Val34, Tyr35, Trp37, Arg104, Leu105, Leu106, Asn108 and Val109, among which residues Val34, Trp37, Leu105 and Leu106 are absolutely conserved, and Tyr35 is replaced by His only in one of four beta-chain isoforms A0A8C6RII4 of *Nannospalax galili*^HR^, and Asn108 is replaced by Asp only in one of three beta-chain isoforms G5BS33 of *Heterocephalus glaber*^HR^ (Appendix A*)*.

Two GSH binding sites (in the oxy form only) are located at the exit line from the hemoglobin tetramer cavity between the two beta subunits and include only beta-chain residues Val1, Leu81, Lys82, Glu101, Phe103, Arg104, Gly136, Ala138, Asn139, Ala140, Ala142, His143 and His146, of which Val1, Leu81, Phe103, Gly136, Ala138, Ala140 and His146 are conserved in all organisms, both normoxia- and hypoxia-resistant, with Lys82 replaced by Asn in both HbB isoforms A0A481WR13 and A0A481WPM8 of *Hydrodamalis gigas*^HR^ and His143 replaced by Ala in all four known HbB isoforms A0A1S3WPY1, P02059, A0A1S3ADW1 and A0A1S3AE42 of *Erinaceus europaeus*^N^.

## 3. Discussion

No clear correlation between hypoxia tolerance and the number of cysteine residues in the alpha- and beta-chains of hemoglobin was found. No single conserved cysteine residue was found in the alpha-chain of hemoglobin (Appendix A). At the same time, a conserved cysteine residue betaCys93 was detected in the beta-chain, which is present in all studied species, both hypoxia-resistant and normoxic mammals, regardless of how many cysteine residues (from 1 to 6, Appendix A) are present in the isoform. HbB subunit cysteine 93 is known to have several different physiological functions, including protection of iron oxidation by peroxide binding [24], binding of endogenic NO and hydrogen sulfide [25,26] and the regulation of dimeric-tetrameric Hb equilibrium [27], and is capable of binding glucose moiety [28]. A perfect review of the role of Cys93 is given by Alayash [25]. Site-directed mutagenesis studies in which βCys93 was replaced by other amino acids (exhibiting different degrees of hydrophobicity) or chemically modified with sulfhydryl reagents unequivocally confirmed the critical role played by this residue in maintaining structural-functional integrity and O_2_ binding properties of Hb [29]. Other cysteine residues in hemoglobin are not as well studied. In vitro studies showed that βCys13 in mouse beta-hemoglobins was more susceptible to disulfide exchange with GSSG than βCys93 [22].

A marker of oxidative stress in erythrocytes is glutathionylation of hemoglobin [30], which not only protects protein thiol groups but also changes the affinity of hemoglobin for oxygen. It is important to note that cysteine 93 is the main glutathionylated residue of human hemoglobin, the glutathionylation of which leads to a 5–6-fold increase in the affinity of hemoglobin for oxygen, reducing the cooperativity of binding [17,18,31]. The appearance of glutathionylated hemoglobin with increased affinity to oxygen leads to the fact that such hemoglobin will not only better bind oxygen in the lungs at lower concentrations of oxygen in the air but also give oxygen at higher concentrations of carbon dioxide and lower concentrations of oxygen in tissues and, therefore, will be able to deliver oxygen to the farthest, most hypoxic tissues. When oxidative stress increases, the percentage of glutathionylated forms of hemoglobin will increase, improving tissue supply. We believe that this is a mechanism of adaptation of normoxia-resistant species to hypoxia conditions. Hypoxia-resistant species, due to this mechanism, can also expand the range of oxygen and carbon dioxide concentrations acceptable for life. Thus, hemoglobins of both hypoxia-resistant and hypoxia-unstable species are capable of glutathionylation by Cys93, which indicates the existence of a single mechanism of redox regulation of hemoglobin function during the development of oxidative stress in hypoxia-resistant and nonresistant species.

Testing of the neotenic theory of hypoxia tolerance [7] has shown that sequences of HbB subunits of some mammalian species, in particular aquatic mammals *Hydrodamalis gigas*^HR^ and *Trichechus manatus latirostris*^HR^, and some HbB isoforms of *Nannospalax galili*^HR^ and *Mus musculus*^N^ are indeed more similar to fetal and embryonic sequences of other mammals than to adult HbB sequences, but this observation is not widespread. In the other hypoxia-tolerant mammalian species considered, the change in hemoglobin affinity for oxygen arose by independent pathways from the adult form of hemoglobin. The adult-type alpha subunit replaces the fetal form at about eight weeks of human fetal age, and the adult version of the beta-chain of hemoglobin completely displaces the fetal forms by about 3.5 months of newborn age [3]. This observation is in line with the fact that adult HbA sequences have no similar fetal or embryonic forms of alpha-like subunits among adult HbA sequences of hypoxia-tolerant animals. In addition, there are no conserved cysteine residues in HbA, indicating that there is no common mechanism of redox regulation of hemoglobin function by cysteines of the alpha subunit.

The ability of oxyhemoglobin to bind glutathione noncovalently and release it upon deoxygenation [16] also plays an important role in protecting erythrocytes from oxidative stress when passing through tissues with low deoxygenation. Formation of a complex with GSH leads to changes in a number of properties of human hemoglobin, including changes in the heme environment, the environment of tryptophan residues, decreased thermostability of the protein [15] and a slight increase in oxygen affinity [16]. The binding sites of glutathione to oxy- and deoxyhemoglobin were prescanned using molecular docking, based on experimental data indicating that the oxy form binds four and the deoxy two GSH molecules [16]. Data on thermodynamic parameters of binding indicate that hydrophobic interactions play an important role in the formation of the complex, which correlates well with the modeling data, which testify to the participation of hydrophobic residues in the formation of the complex [16]. In addition, experimental data on a significant change in tryptophan fluorescence confirmed our data [16] that Trp37 (which makes the main contribution to tryptophan fluorescence of hemoglobin) takes part in the formation of the complex [15]. Conserved glutathione binding sites suggest that the mechanism of glutathione release during deoxygenation is also realized in hypoxia-resistant animals. This may play an important role in case of an abrupt change in oxygen level in the environment, for example, when a burrowing animal goes to the surface, which leads to oxidative stress in tissues. GSH is required to protect red blood cells from such tissue stress. It should be noted that it has been shown on human hemoglobin that complexing with GSH increases the affinity of hemoglobin for oxygen by 15% [16]. The release of glutathione from the complex with hemoglobin, thus, reduces the affinity of hemoglobin to oxygen and promotes the release of oxygen into the tissues of the body.

## 4. Materials and Methods

### 4.1. Hemoglobin Sequences Dataset

Some animal species are more resistant to hypoxia than others. This feature allows them to live and reproduce themselves in high-altitude areas, soil and other environments with a low oxygen concentration. Species with known resistance to hypoxia include *Ailurus fulgens*^HR^ [32], *Bos mutus grunniens*^HR^ [33], *Canis lupus chanco*^HR^ [34], *Capra caucasica caucasica*^HR^, *Capra nubiana*^HR^ [35], *Cavia porcellus*^HR^ [36], *Chinchilla lanigera*^HR^ [37], *Colobus guereza*^HR^ [38], *Ctenomys rionegrensis*^HR^ [39,40], *Heterocephalus glaber*^HR^ [6], *Lama guanicoe*^HR^ [41], *Marmota flaviventris*^HR^ [42], *Ochotona curzoniae*^HR^ [43], *Pantholops hodgsonii*^HR^ [35], *Peromyscus maniculatus*^HR^ [44], *Procavia capensis habessinica*^HR^ [45], *Rhinopithecus roxellana*^HR^ [46], *Spalacopus cyanus*^HR^ [39,40], *Spalax ehrenbergi*^HR^ [8], *Talpa europaea*^HR^ [47], *Tamias umbrinus*^HR^ [48], *Theropithecus gelada*^HR^ [49], *Ursus thibetanus*^HR^ [50] and *Vicugna pacos*^HR^ [51]. We selected all hemoglobin sequences of these species from the Uniprot database (https://www.uniprot.org/, accessed on 20 August 2023). In addition, we took hemoglobin sequences of normoxia species for comparison: *Homo sapience*^N^, *Bos Taurus*^N^, *Sus scrofa*^N^, *Capra hircus*^N^, *Mus musculus*^N^, *Rattus norvegicus*^N^ and *Erinaceus europaeus*^N^. A complete list of the hemoglobin sequences reviewed is provided in the Appendix A. In the text, normoxic species are indicated by an upper “N” after the species name and hypoxia-resistant species by “HR”.

We collected all annotated hemoglobin sequences for all these species (listed in Appendix A) from Uniprot database (www.uniprot.org, accessed on 20 August 2023).

#### 4.1.1. Alpha Hemoglobins

We believe that some of the abovementioned hemoglobin sequences hardly exist in erythrocytes as part of the hemoglobin tetramer in reality and, if they do, they exist only in the form of a monomer and perform another function. For example, the sequences of hemoglobin alpha A0A0A0MP82 and A0A8L2QLW7 of *Rattus norvegicus* have an insertion after the 94th residue (by numbering of human hemoglobin alpha in alignment) of 43 amino acid residues. Structure modeling showed (Appendix A) that an insertion at this position of even a small number of amino acid residues disrupts the normal interaction interface with the beta subunit and an insertion of 43 residues makes complex binding with the beta subunit completely impossible. These sequences were predicted from the rat genome sequence and found as transcripts in the brain [52], but their translation and the presence of a protein product have not been proven. It is also possible that the 43 residue insertion is an intron that should be removed during RNA processing. Such hemoglobin sequences, whose existence in the hemoglobin tetramer seems doubtful, are shown in gray in Appendix A.

The A0A8I5ZYH2 and A0A8I6AWV0 sequences of *Rattus norvegicus* contain an extended insertion at the N-terminus of 45 amino acid residues, which, according to structure-by-homology modeling, would not interfere with interaction with the beta-chain and tetramer association but would interfere with tight packing of hemoglobin in the erythrocyte. Also, the sequences of alpha-hemoglobin of *Sus scrofa* A0A4X1UM57, A0A4X1UMN0 and A0A4X1UMN5 have extended insertions of 88 amino acids at the N-terminus. All of these sequences were excluded from further consideration. Multiple alignments and an evolutionary tree were constructed using the remaining alpha-chain sequences (Appendix A).

#### 4.1.2. Beta Hemoglobins

We excluded the sequences Q9TT33 *Colobus guereza* and E9Q223 *Mus musculus*, which lack significant sequence fragments from the N- and C-termini, respectively, from the sample.

#### 4.1.3. Mu Hemoglobins

We excluded the Sus scrofa sequence A0A5G2QP95 due to long insertion of 50 amino acids after the 31st residue.

### 4.2. Hemoglobin ASA Measurement

Accessible surface area (ASA) was measured with the program MOE 2019.0102 software.

### 4.3. Phylogenetic Analyses

For analyses of conserved residues and phylogenetic relationships, we used UPGMA clustering by similarity using BLOSUM62 matrix by MOE 2019.0102 software.

## 5. Conclusions

Our data suggest that there is a common mechanism of redox regulation of hemoglobin function during oxidative stress in hypoxia-resistant and hypoxia-sensitive mammalian species. This mechanism includes deposition by oxyhemoglobin of reduced glutathione molecules, which are partially released during deoxygenation, as well as glutathionylation of betaCys93 under conditions of oxidative stress to produce hemoglobin with higher affinity to oxygen. This mechanism allows the mammalian organism to adapt to unfavorable conditions and increase the range of oxygen and carbon dioxide concentrations suitable for life.

## Figures and Tables

**Figure 1 ijms-25-00053-f001:**
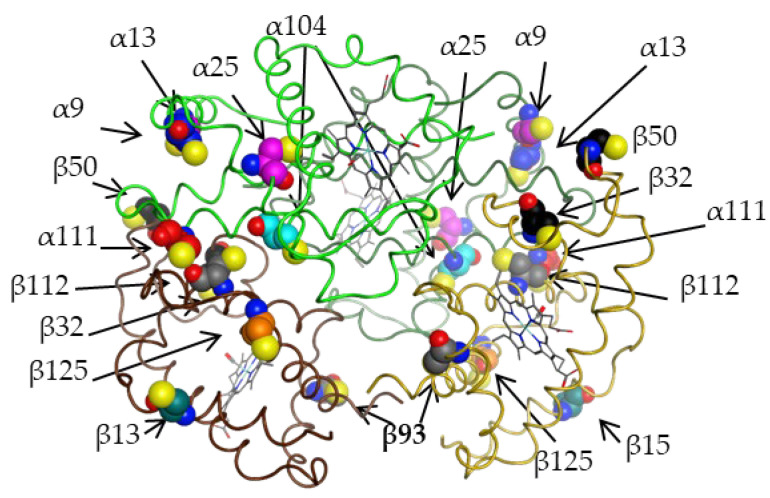
Spatial arrangement of all possible cysteine residues in the structure of hemoglobin tetramer (on the structure of the oxy form of human hemoglobin 1r1x). Alpha-chains are shown with green and dark green lines; beta-chains with brown and light brown lines. The alpha-chains show residues 9, 13, 25, 104 and 111. The beta-chains show residues 13, 32, 50, 93, 112 and 125. Buried and inaccessible to solvent residues are α25 and α104; β32 and β93 are solvent-accessible only in deoxy form; the rest are solvent-accessible and can be glutathionylated both in oxy and deoxy forms.

## Data Availability

Data are contained within the article or Appendix A.

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
