# Peer review of "Glutathione Non-Covalent Binding Sites on Hemoglobin and Major Glutathionylation Target betaCys93 Are Conservative among Both Hypoxia-Sensitive and Hypoxia-Tolerant Mammal Species"

_ijms, 2023, doi:10.3390/ijms25010053_

Round 1

Reviewer 1 Report

Comments and Suggestions for Authors

Dear editor and authors:

In this paperthe number of CysHisArg and Lys of different isoforms of hemoglobin from different hypoxia-tolerant and hypoxia-sensitive mammals were counted. And classified Hemoglobin into Alpha-Like and Beta-Like by Phylogenetic tree. The amino acid of Alpha-Like and Beta-Like hemoglobin were compared to analyze the distribution of Cys in different isoforms, whether it is conservative, whether it is solvent accessible.The conservatism of Cys94 in beta chain and its effect on glutathione hemoglobin oxygen affinity were illustrated by citing literatures. It provides the information basis and experimental basis for the follow-up researchers to modify hemoglobin.

There are some questions about this article

1)In Line 138-141, mentioned“The most dissimilar sequences are A0A481WPK4A0A8C2R2Z3G5BXY4”, while in Figure S1 A0A481WPK4 was classified into Beta-Like hemoglobin, Is there something wrong with such an expression,Because this may be caused by the size of your database.

2)In discussion, the result not from this study are used. For example, in line 232 to 234,you have used your previous work to testify against this work, but there is no obvious causal relationship between this and your current work, you should explain in more detail.

3)In line 2.2.1 HbA,you mentioned except Cys105 and Cys26 are not solvent accessible, the remain residues are accessible. So, What is the role of Cys105? because it is conserved in most HbA.

4)Can you explain the role of Cys14 in HbA?  Because Cys14 is the only common residue of solvent accessible in HbA    

5)In line 336 because of what?  Can you complete this sentence?

6)Your title is prove that Cys94 is conservative, while your discussion is not revolve around the title.  

Comments on the Quality of English Language

Dear editor and authors:

In this paperthe number of CysHisArg and Lys of different isoforms of hemoglobin from different hypoxia-tolerant and hypoxia-sensitive mammals were counted. And classified Hemoglobin into Alpha-Like and Beta-Like by Phylogenetic tree. The amino acid of Alpha-Like and Beta-Like hemoglobin were compared to analyze the distribution of Cys in different isoforms, whether it is conservative, whether it is solvent accessible.The conservatism of Cys94 in beta chain and its effect on glutathione hemoglobin oxygen affinity were illustrated by citing literatures. It provides the information basis and experimental basis for the follow-up researchers to modify hemoglobin.

There are some questions about this article

1)In Line 138-141, mentioned“The most dissimilar sequences are A0A481WPK4A0A8C2R2Z3G5BXY4”, while in Figure S1 A0A481WPK4 was classified into Beta-Like hemoglobin, Is there something wrong with such an expression,Because this may be caused by the size of your database.

2)In discussion, the result not from this study are used. For example, in line 232 to 234,you have used your previous work to testify against this work, but there is no obvious causal relationship between this and your current work, you should explain in more detail.

3)In line 2.2.1 HbA,you mentioned except Cys105 and Cys26 are not solvent accessible, the remain residues are accessible. So, What is the role of Cys105? because it is conserved in most HbA.

4)Can you explain the role of Cys14 in HbA?  Because Cys14 is the only common residue of solvent accessible in HbA    

5)In line 336 because of what?  Can you complete this sentence?

6)Your title is prove that Cys94 is conservative, while your discussion is not revolve around the title.  

Author Response

The authors thank the reviewers for their careful reading of the article and valuable comments. We have taken them into account. We highlighted all changes in yellow in the revised version of the manuscript. In addition, we brought the numbering of amino acid residues from the Uniprot numbering to the generally accepted one used in all publications about hemoglobin. These numberings differ by 1 because the first methionine residue in both the alpha and beta subunits is removed during post-translational processing.

Reviewer 1

In this paperthe number of CysHisArg and Lys of different isoforms of hemoglobin from different hypoxia-tolerant and hypoxia-sensitive mammals were counted. And classified Hemoglobin into Alpha-Like and Beta-Like by Phylogenetic tree. The amino acid of Alpha-Like and Beta-Like hemoglobin were compared to analyze the distribution of Cys in different isoforms, whether it is conservative, whether it is solvent accessible.The conservatism of Cys94 in beta chain and its effect on glutathione hemoglobin oxygen affinity were illustrated by citing literatures. It provides the information basis and experimental basis for the follow-up researchers to modify hemoglobin.

There are some questions about this article

1In Line 138-141, mentioned “The most dissimilar sequences are A0A481WPK4A0A8C2R2Z3G5BXY4”, while in Figure S1 A0A481WPK4 was classified into Beta-Like hemoglobin, Is there something wrong with such an expression,Because this may be caused by the size of your database.

Thanks for the note, we have fixed it.

2In discussion, the result not from this study are used. For example, in line 232 to 234,you have used your previous work to testify against this work, but there is no obvious causal relationship between this and your current work, you should explain in more detail.

Corrected

3In line 2.2.1 HbA,you mentioned except Cys105 and Cys26 are not solvent accessible, the remain residues are accessible. So, What is the role of Cys105? because it is conserved in most HbA.

Cysteine 104 of the alpha subunit, in both oxy and deoxy forms, contacts with Gln127 of the beta subunit, so we believe it is important for maintaining the stability of the alpha/beta subunit interface. A description of the role of HbA Cys104 has been added to the results section.

4)Can you explain the role of Cys14 in HbA?  Because Cys14 is the only common residue of solvent accessible in HbA  

In vitro studies showed that β Cys13 in mouse HbbD beta globins was more susceptible to disulfide exchange with GSSG than β Cys93 (Hempe, Ory-Ascani, and Hsia 2007). Also β Cys13 is able to bind dimethylarsinous Acid in rat blood (Lu et al. 2007). A description of the role of alphaCys13 has been added to the results section. 

5In line 336 because of what?  Can you complete this sentence?

Thank you for the comment, we removed this section, it’s a technical error.

6Your title is prove that Cys94 is conservative, while your discussion is not revolve around the title.  

We took this comment into account and focused the discussion on the conservation of the binding site for glutathione and conservation of Cys93.

Reviewer 2 Report

Comments and Suggestions for Authors

This manuscript is not fully convincing. One of the main aspects concerns the involvement of cysteine residues in the binding of glutathione. Cysteines are known to have several different physiological functions in hemoglobins, including protection of iron oxidation, dimerization processes, binding of NO and hydrogen sulfide. This overview is lacking completely. The authors need to desrcibe these different mechanisms that influence the properties of the protein. To what extent are the proteins "loaded" with glutathione? Much data are presented in the supplementary material, but are not really discussed in the manuscript. I am lacking scientific depth and more experimental data should be included to support the modelling work.

Comments on the Quality of English Language

There are some minor sections where paragraphs appear to have been deleted by mistake.

Author Response

The authors thank the reviewers for their careful reading of the article and valuable comments. We have taken them into account. We highlighted all changes in yellow in the revised version of the manuscript. In addition, we brought the numbering of amino acid residues from the Uniprot numbering to the generally accepted one used in all publications about hemoglobin. These numberings differ by 1 because the first methionine residue in both the alpha and beta subunits is removed during post-translational processing.

Reviewer 2

This manuscript is not fully convincing. One of the main aspects concerns the involvement of cysteine residues in the binding of glutathione. Cysteines are known to have several different physiological functions in hemoglobins, including protection of iron oxidation, dimerization processes, binding of NO and hydrogen sulfide. This overview is lacking completely. The authors need to desrcibe these different mechanisms that influence the properties of the protein.

Thanks for your comment, we tried to clarify the main ideas and added a description of the role of modifications of cysteine residues in hemoglobin in the discussion.

To what extent are the proteins "loaded" with glutathione?  

Hemoglobin can form a non-covalent complex with glutathione, and the number of glutathione molecules depends on the degree of oxygenation of hemoglobin. It was found (Fenk et al. 2022), that free GSH concentration in fully oxygenated RBCs translates from an average of 2.5 μmol/g Hb (Fig. 2A) to 1.23 mmol/L cell water. Decrease in hemoglobin oxygen saturation from 98 to 20% (Fig. 2A) resulted in an increase in free GSH up to 6 μmol/g Hb which translates to 2.96 mmol/L cell water. Maximal amount of GSH that could be released by 5.23 mM oxy-Hb makes up 10.5 mM (two GSH molecules per 1 Hb molecule). So, cytosolic Hb concentration (5.4 mM) is comparable with that of GSH (1.2–3.5 mM). During deoxygenation, two of the four bound glutathione molecules dissociate from the hemoglobin complex. The oxyform of hemoglobin can associate with a maximum of 4 GSH, and deoxyhemoglobin can associate a maximum of 2 GSH. The non-covalent complex we discovered is described in detail in (Fenk et al. 2022).

Hemoglobin can also bind glutathione covalently, through the SS bridge, which leads to the formation of glutathionylated protein. Currently, glutathionylation of Cys93 has been proven, and glutathionylation of Cys112 has also been suggested. That is, from 2 to 4 glutathione molecules can be covalently bound to hemoglobin. Normally, about 4% of hemoglobin is glutathionylated in erythrocytes, and in pathologies accompanied by oxidative stress this value can increase significantly up to 69%(Mieyal et al. 2008). The properties of the protein during noncovalent and covalent binding of glutathione differ significantly and are described by us in the (Kuleshova et al. 2023) work.

Fenk, Simone et al. 2022. “Hemoglobin Is an Oxygen-Dependent Glutathione Buffer Adapting the Intracellular Reduced Glutathione Levels to Oxygen Availability.” Redox Biology 58: 102535.

Hempe, James M., Jeannine Ory-Ascani, and Daniel Hsia. 2007. “Genetic Variation in Mouse Beta Globin Cysteine Content Modifies Glutathione Metabolism: Implications for the Use of Mouse Models.” Experimental Biology and Medicine 232(3): 437–44.

Kuleshova, Iuliia D. et al. 2023. “Changes in Hemoglobin Properties in Complex with Glutathione and after Glutathionylation.” International Journal of Molecular Sciences 24(17): 13557.

Lu, Meiling et al. 2007. “Binding of Dimethylarsinous Acid to Cys-13α of Rat Hemoglobin Is Responsible for the Retention of Arsenic in Rat Blood.” Chemical Research in Toxicology 20(1): 27–37.

Mieyal, John J. et al. 2008. “Molecular Mechanisms and Clinical Implications of Reversible Protein S-Glutathionylation.” Antioxidants & Redox Signaling 10(11): 1941–88.

Much data are presented in the supplementary material, but are not really discussed in the manuscript.

Supplementary material contains a phylogenetic tree of hemoglobin sequences, accessible amino acid surface area of human hemoglobin, and hemoglobin alpha and beta chain alignments to determine conservation. These results are used in different parts of the article. In addition, we believe that supplementary material may have independent value as reference information for hemoglobin researchers.

 I am lacking scientific depth and more experimental data should be included to support the modelling work.

This is a theoretical work that is based on previously obtained experimental data.

There are some minor sections where paragraphs appear to have been deleted by mistake.

Corrected

Round 2

Reviewer 1 Report

Comments and Suggestions for Authors

The revised version is suitable for publication